# Associations between the Global Diet Quality Score and risk of type 2 diabetes: Tehran lipid and glucose study

Shahrzad Daei[1�], Firoozeh Hosseini-Esfahani[1☸], Azam Ildarabadi[1], Parvin Mirmiran[1,2]*, Fereidoun Azizi[3]

**1** Nutrition and Endocrine Research Center, Research Institute for Endocrine Sciences, Shahid Beheshti University of Medical Sciences, Tehran, Iran, **2** Department of Clinical Nutrition and Dietetics, Faculty of Nutrition Sciences and Food Technology, National Nutrition and Food Technology Research Institute, Shahid Beheshti University of Medical Sciences, Tehran, Iran, **3** Endocrine Research Center, Research Institute for Endocrine Sciences, Shahid Beheshti University of Medical Sciences, Tehran, Iran

☸ These authors contributed equally to this work.
* parvin.mirmiran@sbmu.ac.ir, Parvin.mirmiran@gmail.com

## Abstract

### Background

Previous studies reported that focusing on healthy lifestyle, especially high diet quality is necessary for preventing type 2 diabetes (T2D). This study investigated the association between the innovative index, the Global Diet Quality Score (GDQS), and the risk of Type 2 Diabetes incidence.

### Methods

In this secondary analysis, we included elective adult participants (n = 5948) from the third and fourth survey of the Tehran Lipid and Glucose Study. Participants checked out until the sixth phase with an average follow-up of 6.65 years. Expert nutritionists collected dietary data using a valid and reliable semi-quantitative food frequency questionnaire. The GDQS were calculated, including healthy and unhealthy food group scores. Biochemical and anthropometric characteristics were assessed during the first and follow-up surveys. Multi-variable Cox proportional hazards regression models were used to estimate the progression of T2D in association with the GDQS.

### Results

This study was implemented on 2,688 men and 3,260 women, respectively with the mean (SD) age of 41.5(14.1) and 39.3(13.02) years. A total of 524 subjects were found to have had T2D incidence. The healthy component of GDQS was conversely associated with T2D incidence [HR: 1, 0.91 (0.84–0.98), 0.91 (0.84–0.98), 0.84 (0.77–0.92) P trend = <0.001] in an adjusted model. The unhealthy component of GDQS was conversely associated with T2D incidence in an adjusted model [HR: 1, 0.86 (0.80–0.92), 0.93 (0.86–1.01), 0.89 (0.81–0.98) P trend = 0.009].

**Data Availability Statement:** The datasets generated and analyzed during the current study are not publicly available because data contains

sensitive individual information and data are owned by Research Institute for Endocrine Sciences, Shahid Beheshti University of Medical Sciences. Tehran, Iran Data are available from the ethics committee of the Research Institute for Endocrine Sciences, Shahid Beheshti University of Medical Sciences, Tehran, Iran, whenever data request has been sent. No.24, Arabi Street, Yemen Avenue, Chamran Highway Fax: +98 (21)22402463 Postal code: 1985717413 Email: info@endocrine.ac.ir.

**Funding:** This work was supported by the Research Institute for Endocrine Science, Shahid Beheshti University of Medical Science (Tehran, Iran) under Grant number 43003733. The funders had no role in study design, data collection and analysis, decision to publish, or preparation of the manuscript.

**Competing interests:** The authors have declared that no competing. interests exist.

## Conclusion

The results of this study suggested that higher adherence to the healthy component of GDQS and lower intake of the unhealthy component decreased the risk of T2D incidence.

## Introduction

Diabetes type 2 (T2D) is on the rise all over the world [1]. Approximately 422 million adults worldwide had T2D in 2014, by 2045, the number is estimated to be 629 million [2]. In the Iranian population, there are >800,000 new cases of T2D per year, with a T2D incidence rate of 36.3 per 1000 person-years [3]. To reduce T2D prevalence, focusing more on identifying risk factors is necessary. Therefore, prohibition through a healthy lifestyle, especially high dietary quality, is a pivotal approach [4]. Diet has a crucial role in the development of T2D [1]. Current meta-analysis study suggest that the Mediterranean diet, the dietary approaches to stop hypertension (DASH), and the alternative healthy eating index (AHEI) are conceivable dietary patterns to prevent diabetes [5]. The Global Diet Quality Score (GDQS) is a unique and practical index for evaluating dietary quality worldwide. This simple and inexpensive score is suitable for comparing populations with different economic statuses [4]. It applies to follow over time and surveillance systems and program monitoring [6]. The GDQS comprises 25 food groups that contribute significantly to nutrient intake and the risk of non-communicable diseases (NCDs) [6]. Food groups are divided into 16 healthy including kinds of fruits and vegetables, legumes, nuts and seeds, whole grain, fish, poultry, liquid oil, low-fat dairy, and egg; 7 unhealthy including processed meats, refined grains, sweets, sugar-sweetened beverages, potato or cassava flour, juice, and deep fried foods; and two unhealthy when consumed more than recommendation including red meat and high-fat dairy. Based on the results of previous reports, the GDQS was positively associated with body mass index (BMI) among pregnant women in rural Ethiopia [7] and inversely associated with metabolic syndrome and nutrient inadequacy in Chinese adults [8]. Evidence from prospective studies showed that higher coherence to GDQS was associated with a lower risk of T2D incidence in US women and a lower in weight and waist circumference (WC) increase in Mexican women [9]. Insight of increasing the incidence of T2D in Iran and the important role of diet in the prevention of this disease, we investigated the association between the GDQS, its healthy and unhealthy food group components and the risk of T2D incidence in a group of Tehranian adults.

## Methods

### Study design and subjects

The Tehran lipid and glucose study (TLGS) was performed prospectively on the 13th district of Tehran (the capital of Iran) residents to identify risk factors for non-communicable diseases [10, 11]. According to this study, the initial sample of 15005 participants (aged ≥3 years) were enrolled between 1999 and 2001. During the study, participants were followed every three years: wave 2 (2002–2005), wave 3 (2005–2008), wave 4 (2008–2011), wave 5 (2012–2015), wave 6 (2015–2018) to amend their demographic and health-related data, as well as identified diseases that have recently emerged. For this secondary analysis of subjects participating in waves 3 and 4 (baseline of our study), 8048 individuals aged ≥18 years were randomly selected to complete the dietary assessment based on age and sex distribution. Subjects with over- or under-reporting of energy intake (≥4,200 or <800 kcal/day) (n = 780) were excluded [12],

and a total of 7268 individuals with accessible biochemical, anthropometric, and dietary data were entered as the baseline population. They were tracked until phase 6; subjects who entered phase 3 or 4 in this study were respectively followed three and twice for the outcome measurement. Among these subjects, pregnant or lactating women, participants with T2D diagnosis based on fasting blood glucose (FBG) measurements or self-reported use of glucose-lowering medications at baseline (n = 597), and subjects with missing data (n = 208) were excluded. Also, subjects who did not provide follow-up data were excluded (n = 515) from the study participants. As a result, 5948 subjects remained for analysis (Fig 1).

## Dietary intake measurements

A valid and reliable 147-item semi-quantitative food frequency questionnaire (FFQ) was applied to assess dietary intakes [12, 13]. At each follow-up visit, a trained nutritionist asked participants to report how often specific food items were consumed over the previous year in terms of daily, weekly, or monthly consumption. The portions of taken foods were changed from household measurements to grams per day. Due to the incompleteness and limited data on the nutrient content of cooked food items in the Iranian Food Composition Table (FCT), United States Department of Agriculture (USDA) data was used. The Iranian FCT was applied for national foods that could not be incorporated into the USDA portion size [2]. Also, the nutrient contents of food items and food groups were estimated cumulatively during follow-up visits from the first survey regarding the period of diabetes diagnosis or the last follow-up survey.

Globally, the GDQS is a validated index that reflects nutritional adequacy and predicts major NCDs [6]. Scores are designated subject to 3 or 4 categories of eaten amounts (gr/day) of specific food groups. It comprises 16 healthy food groups (all kinds of fruits and vegetables, nuts and seeds, legumes, whole grains, oils, fish and poultry, low-fat dairy and egg), seven unhealthy food groups (processed meat, refined grains, and baked goods, sweets, sugar-sweetened beverages, packaged fruit juices, deep fried foods) and two unhealthy food group when consumed in the excessive amounts (high-fat dairy products and red meat) (S1 Table).

In the healthy food group, higher scores were given for higher intakes (0.5–4 points); in the unhealthy group, higher scores were given for lower intakes (0–2 points); and in the unhealthy in the excessive amount group, higher scores are given until definite amounts have been eaten, after which no score is donated (0–2 points). The range of the GDQS is from 0 to 49.

## Physical activity measurements

A trained interviewer used a Persian-translated modifiable activity questionnaire (MAQ) to estimate physical activity levels. The questionnaire has high reliability and moderate validity, according to a previous study [14]. The time, frequency, and intensity of light, middle, high, and challenging activities were recorded during the last year based on routine daily activities. The activity data were transformed into metabolic equivalent/minutes/week (Met/min/week) [15].

## Blood pressure and anthropometric measurements

Weight (kg) was sized with a digital scale (model 707, range 0.1–150 kg; Seca, Hamburg, Germany) and light clothes with no shoes in the standing position. Height was measured with a non-flexible tape measure (model 208 Portable Body Meter Measuring Device; Seca, accuracy 0.5 cm) in the standing position with their head in the Frankfort horizontal plane and without shoes. We measured WC after exhaling without pressure on the surface of the body with the light clothing. Our measurements were accurate to 0.1 centimeters. A standardized mercury

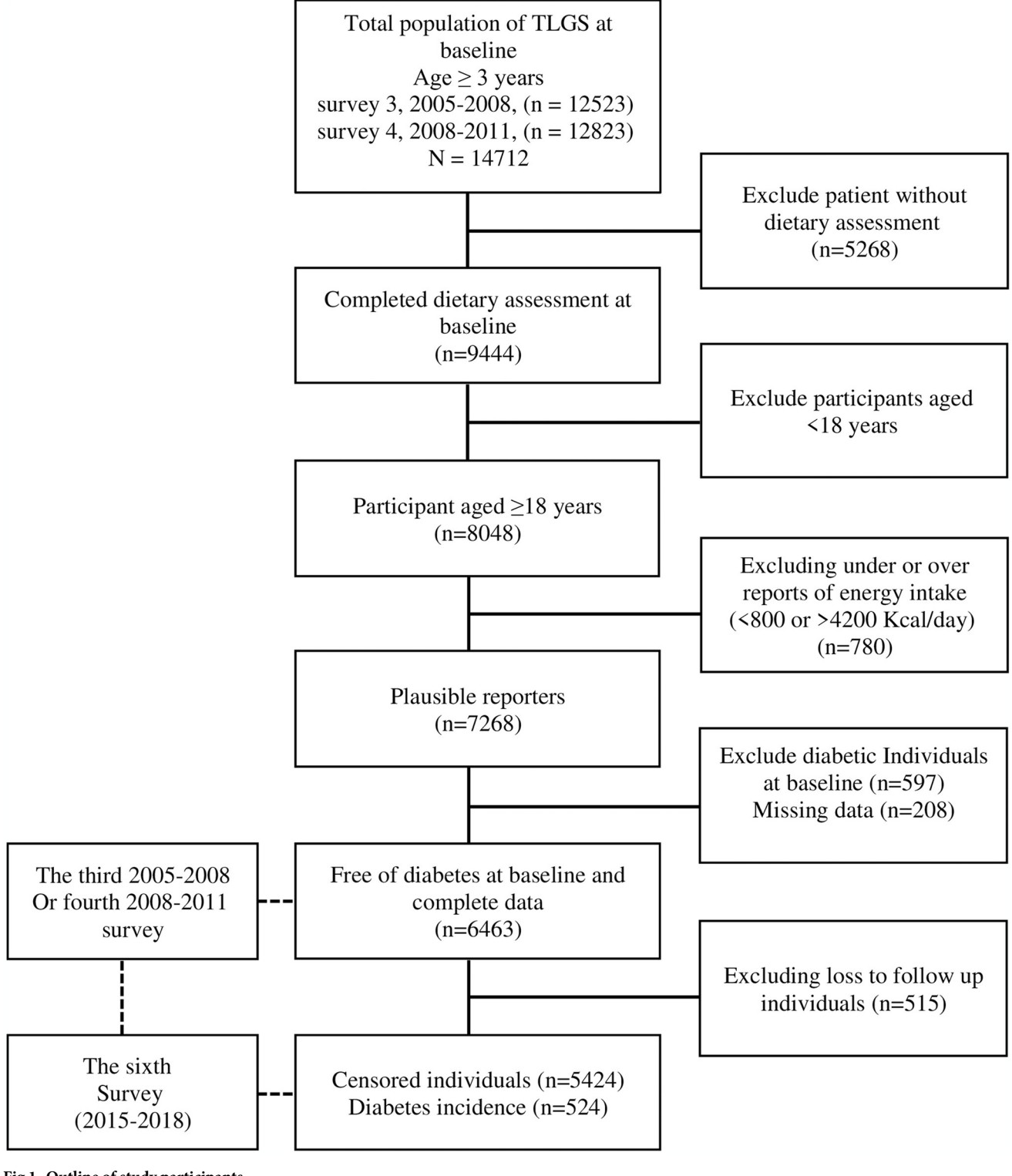

**Fig 1. Outline of study participants.**

sphygmomanometer was applied to determine the systolic and diastolic blood pressure (SBP and DBP) (mmHg) according to accepted protocols, which were explained previously [10].

## Biochemical analysis

Blood samples were collected between 7:00 and 9:00 a.m., after 12–14 hours of overnight fasting. The technician analyzes blood samples on the day of the blood collection using the Selectra 2 auto-analyzer at the TLGS research laboratory. They computed FBG concentration using the enzymatic colorimetric and glucose oxidase techniques (Vital Scientific, Spankeren, the Netherlands). In each visit, all participants who did not take glucose-lowering medications were given 82.5 g glucose monohydrate solution (equivalent to 75 g anhydrous glucose) orally to measure 2 hours of postprandial glucose.

Triglycerides (TG) and total cholesterol (TC) concentrations were estimated by an enzymatic colorimetric method using glycerol phosphate oxidase and cholesterol esterase and cholesterol oxidase, respectively (Pars Azmoon Inc, Tehran, Iran). High-density lipoprotein cholesterol (HDL-C) concentration was evaluated after precipitation of the apolipoprotein B-containing lipoproteins with phosphotungstic acid. The inter- and intra-assay coefficients of variations of glucose were 2.2%. The inter- and intra-assay coefficients of variations of TG were 1.6 and 0.6%, respectively [16].

## Diabetes risk score

The diabetes risk score (DRS) was calculated in this way to mitigate confounding factor numbers in the models: family history of T2D (5 points) (a positive family history of diabetes is defined as having at least one parent or sibling with diabetes), TG/HDL-C: <3.5 (0 points), ≥ 3.5 (3 points); waist-to-height ratio: <0.54 (0 points), 0.54–0.59 (6 points), ≥0.59 (11 points), FBG (mg/dl): < 90.09 (0 points), 90.09–99.10 (12 points), 100.90–124.32 (33 points) SBP (mmHg) <120 (0 points), 120–140 (3 points) SBP ≥ 140 (7 points) [17].

## Outcome definition

T2D was defined as fasting plasma glucose levels of ≥126 mg/dl, 2-hour plasma glucose levels of ≥ 200 mg/dl, or self-reported taking glucose-lowering drugs (oral medication or insulin injections) [18].

## Statistical analysis

The IBM SPSS software, version 26, was applied for data analysis. A two-sided P-value <0.05 was recognized as statistically significant. To compare the mean and frequency of participants' baseline characteristics across quartiles of the GDQS, χ2 test and one-way ANOVA were used for categorical and continuous variables, respectively. P for trend across the GDQS subgroup categories was conducted by designating continuous variables in a linear regression model. Multivariable Cox proportional hazard regression analyses were done to assess the hazard ratio (HR) and 95% confidence interval (CI) for T2D incidence across quartiles of the GDQS and the score of healthy and unhealthy food groups. There were no interactions between the GDQS score and age or sex in relation to T2D incidence. The first quartile was considered as the reference. The confounders were chosen according to the literature including, age, sex, education levels (>14 and ≤14 years), diabetes risk score, physical activity (MET/min/week), total energy (kcal/day), and saturated fat intakes (percentage of energy) and smoking (never smoked, past smoked, and current smoker). Also, we applied each confounder in the univariable Cox regression model; A two-tailed P-value <0.2 was used for specifying admission in the

model. The quantitative score of the GDQS was used as a continuous variable to estimate the P for a trend in the Cox proportional hazard regression models. The definition of time to event was based on the mid-time between baseline and the event date (for incidence cases) or the time between baseline and last follow-up (for censored subjects), whichever occurred first.

Two models were constructed: The first one was adjusted for age and sex, and the second one was additionally adjusted for education levels, diabetes risk score, physical activity, total energy intake, saturated fat intake, fiber, and smoking. The proportional hazard assumption was confirmed by the Schoenfeld residuals test and plot of log [-log (survival)] vs. log (time) to see if they are parallel.

The Research Institute for Endocrine Sciences ethics committee, Shahid Beheshti University of Medical Sciences approved the study protocol (IR.SBMU.ENDOCRINE.REC.1401.122) (Grant no. 43003733). All subjects provided written informed consent. It conforms to the provisions of the Declaration of Helsinki.

## Results

Our analysis involved a total of 5948 subjects, comprising 2688 men (% 45.2) and 3260 women (%54.8), aged 41.5(14.1) and 39.3(13.0) years, respectively. Among these subjects, there were 524 new cases of T2D and 5424 censored individuals; the mean follow-up period was 6.63 years. In Table 1, we show the baseline attribute of subjects beyond the quartiles of the GDQS. Subjects in the higher quartiles of GDQS were older than subjects in the lower quartiles

**Table 1. Baseline characteristics of adult participants of the Tehran lipid and glucose study according to the Global Diet Quality Score (GDQS).**

| variables | Quartiles of GDQS | | | | P-value |
| --- | --- | --- | --- | --- | --- |
| | Q1 | Q2 | Q3 | Q4 | |
| | (<26.26) | (26.26–29) | (29.01–13.75) | >13.75 | |
| Baseline age (years) | 37.9 (13.8) * | 39.3 (13.6) * | 41.2(13.4) | 42.5(13.1) | <0.001 |
| Sex (% women) | 780 (53.1) | 818(55.5) | 869 (54.5) | 793(56.2) | 0.359 |
| Smoking | | | | | <0.001 |
| Not smoker | 954 (65) | 1011 (68.8) | 1148 (72.2) | 1066 (75.8) | |
| Ex- smoker [a] | 89 (6.1) | 104 (7.1) | 113(7.1) | 120 (8.5) | |
| Current smoker | 424 (28.9) | 354 (24.1) | 330 (20.7) | 221(15.7) | |
| Education: | | | | | 0.03 |
| Elementary | 206 (14.1) | 188(12.8) | 231(14.5) | 192 (13.7) | |
| Diploma | 908 (62) | 895 (60.9) | 913 (57.4) | 807 (57.4) | |
| Academic | 351 (24) | 386 (26.3) | 447 (28.1) | 406 (28.9) | |
| BMI (kg/m2) | 26.4 (4.9)* | 27.0 (4.8) * | 27.2 (4.6) | 27.5 (4.7) | <0.001 |
| Physical activity (MET/min/week) | 459 (766.9)* | 536 (794) | 574.8 (820.3) * | 659 (934.2) | <0.001 |
| Waist-to-hip ratio | 0.9 (0.09) | 0.9 (0.09) | 0.9 (0.09) * | 0.9 (0.08) * | 0.001 |
| SBP (mm Hg) | 110.7 (15.8) * | 111.3 (16.1) * | 112.9 (16.1) | 113 (16.2) | <0.001 |
| DBP (mm Hg) | 73.7(10.6) | 73.9 (11.1) | 75.1 (10.6)* | 75.3 (10.5)* | <0.001 |
| Total cholesterol (mg/dL) | 182 (37.3) * | 183.6 (37.3) | 186.3 (38.2) | 189.2 (39.1) * | <0.001 |
| TG/HDL ratio | 3.24(2.4) | 3.43 (2.8) | 3.43(2.6) | 3.46(2.9) | 0.15 |
| FBG (mg/dL) | 89.8 (8.9) | 89.9 (8.7) | 90.7 (9.2) * | 91.1 (9.3) * | <0.001 |
| Family history of diabetes (%Yes) | 216 (14.7) | 193 (13.1) | 204 (12.8) | 207 (14.7) | 0.01 |

Values are mean (SD) unless otherwise listed. P values were derived from the analysis of variance and the Chi-square test was used for continuous and dichotomous variables, respectively.

[a]: An individual who has given up cigarette and/or tobacco smoking.

*: P<0.05 for post hoc analysis.

**Table 2. Dietary intake and the score of participants based on quartiles of the Global Diet Quality Score (GDQS).**

| variable | Quartiles of GDQS | | | | p-value |
|---|---|---|---|---|---|
| | Q1 | Q2 | Q3 | Q4 | |
| Energy intake, kcal/d | 1998(572.9) | 2250 (577.8) | 2353 (547.3) | 2578 (562.9) | <0.001 |
| Carbohydrate (% of energy) | 58.7 (6.2) | 59 (6.05) | 59.2 (5.3) | 59.3 (4.7) | 0.001 |
| Protein (% of energy) | 14 (2.8) | 14.7 (3.3) | 15 (2.2) | 15.4 (2.2) | <0.001 |
| Total fat (% of energy) | 30.1 (6.1) | 29.9 (7.1) | 29.5 (4.8) | 29.4 (5.7) | <0.001 |
| SFA (% of energy) | 10 (3.8) | 9.9 (5.4) | 9.5 (2.2) | 9.2 (1.8) | <0.001 |
| PUFA (% of energy) | 6.2 (1.9) | 6.1 (5.2) | 5.9 (1.5) | 6 (1.4) | 0.05 |
| MUFA (% of energy) | 10.3 (2.6) | 10.2 (5.4) | 9.9 (1.9) | 9.7 (1.9) | <0.001 |
| GDQS score [b] | 23.6 (2.2) | 27.8 (0.8) | 30.5 (0.81) | 34.2 (1.7) | <0.001 |
| Healthy GDQS components [c] | 13.4 (2.8) | 17.4 (1.9) | 19.8 (1.9) | 23.3 (2.1) | <0.001 |
| Unhealthy GDQS components [d] | 7.9(1.9) | 7.8 (1.8) | 8.1 (1.7) | 8.2 (1.7) | <0.001 |
| Unhealthy in excessive amounts [e] | 2.2 (0.8) | 2.50 (0.6) | 2.6 (0.6) | 2.7 (0.5) | <0.001 |

Values are mean (SD). P values were derived from the analysis of variance.

[b] GDQS score components: Healthy + Unhealthy + Unhealthy in excessive amounts.

[c] Food groups of healthy components of GDQS include Citrus fruits, Deep orange fruits, Other fruits, Dark green leafy vegetables, Cruciferous vegetables, Deep orange vegetables, Other vegetables, Legumes, Deep orange tubers, Nuts and seeds, Whole grains, Liquid oils, Fish and shellfish, Poultry and game meat, Low-fat dairy, Eggs (This component was categorized based on quartiles of this score).

[d] Food groups of unhealthy components of GDQS include Processed meat, Refined grains, and baked goods, Sweets and ice cream, Sugar-sweetened beverages, juice, White roots, and tubers, Purchase deep fried foods (This component was categorized based on quartiles of this score).

[e] Food groups of unhealthy excessive amount components of GDQS include High-fat dairy, Red meat.

(P<0.001). There was no statistically significant association between sex and GDQS quartiles. The percentage of current smokers was lower in the upper quartiles of GDQS than in the lower quartiles (P<0.001). In addition, individuals with a higher level of physical activity, BMI, WC, and FBG were in the higher quartiles than in the lower quartiles (P<0.001), but there was no association between TG/HDL ratio and GDQS quartiles.

Furthermore, participants in the higher quartiles of GDQS had higher total protein, carbohydrate, and energy intake and lower total fat, saturated fat (SFA), and monounsaturated fat (MUFA) intakes compared with lower quartiles (P<0.001). Individuals with higher GDQS, healthy, unhealthy, and unhealthy in excessive amount scores were in the upper quartiles of GDQS (In the two groups of unhealthily and unhealthy in excessive amount, scores were reversed, so higher scores indicate a lower intake in these two groups (Table 2).

HRs (95% CI) of T2D for quartiles of the GDQS and its subgroups are shown in Table 3. The incidence of T2D was conversely associated with quartiles of GDQS [HR: 1, 0.91 (0.84–0.98), 0.87 (0.80–0.94), 0.83 (0.76–0.92), P trend <0.001] after adjusting for confounding variable.

The score of healthy food groups of the GDQS was not associated with T2D incidence in the crude model, but after adjusting for confounding variables, the healthy subgroup of GDQS was conversely associated with the T2D incidence [HR: 1, 0.91 (0.84–0.98), 0.91 (0.84–0.98), 0.84 (0.77–0.92) P trend <0.001].

Fig 2 shows Cox proportional hazard regression plot for T2D according to quartiles of healthy component food group of the global diet quality score.

The score of unhealthy food groups of the GDQS [HR: 1, 0.94 (0.88–1.01), 0.92 (0.86–0.99), 0.97 (0.91–1.05) P trend = 0.02] were associated with the incidence of T2D in the crude model; however, our result showed no relationship after adjusting for confounding factors [HR: 1, 0.96 (0.89–1.03), 0.95 (0.89–1.03), 1 (0.92.-1.09) P trend = 0.19]. The score of unhealthy in excessive

**Table 3. HRs (95% CI) of diabetes incidence across categories of the Global Diet Quality Score (GDQS) and its components in adult participants of the TLGS Study (n = 5948).**

| Variables | | Quartiles | | | | P trend |
|---|---|---|---|---|---|---|
| | | Q1 | Q2 | Q3 | Q4 | |
| Incidence diabetes (n) | | 113 | 110 | 153 | 148 | |
| Score | | **12.25–26.25** | **26.50–29** | **29.25–31.75** | **32–41.25** | |
| GDQS [a] | Crude* | Ref | 0.95 (0.89–1.03) | 0.95 (0.89–1.02) | 0.95 (0.88–1.03) | **0.05** |
| | Model adjusted[‡] | Ref | 0.91 (0.84–0.98) | 0.87 (0.80–0.94) | 0.83 (0.76–0.92) | **<0.001** |
| Score | | **≤ 15.75** | **15.76–18.50** | **18.51–21.50** | **≥ 21.51** | |
| Healthy component of GDQS [b] | Crude* | Ref | 0.96 (0.90–1.04) | 1.01 (0.94–1.08) | 0.97 (0.90–1.04) | 0.63 |
| | Model adjusted[‡] | Ref | 0.91 (0.84–0.98) | 0.91 (0.84–0.98) | 0.84 (0.77–0.92) | **<0.001** |
| Score | | **≤ 7** | **7.01–8** | **8.01–9** | **≥ 9.01** | |
| Unhealthy components of GDQS [c] | Crude* | Ref | 0.94 (0.88–1.01) | 0.92(0.86–0.99) | 0.97 (0.91–1.05) | **0.02** |
| | Model adjusted[‡] | Ref | 0.96 (0.89–1.03) | 0.95 (0.89–1.03) | 1.00 (0.92–1.09) | 0.19 |
| Score [d] | | **0–2** | | **3** | | |
| Unhealthy in excessive amount component of GDQS [e] | Crude* | Ref | | 0.89 (0.85–0.94) | | **<0.001** |
| | Model adjusted[‡] | Ref | | 0.89 (0.84–0.94) | | **0.001** |
| score | | **≤ 9** | **9.01–11** | **11.01–12** | **≥ 12.01** | |
| Total unhealthy score [f] | Crude* | Ref | 0.85 (0.80–0.91) | 0.89 (0.82–0.96) | 0.87 (0.80–0.96) | **<0.001** |
| | Model adjusted[‡] | Ref | 0.86 (0.80–0.92) | 0.93(0.86–1.01) | 0.89 (0.81–0.98) | **0.009** |

Test for trend was performed based on the GDQS and its components as a continuous variable in the adjusted Cox proportional hazard model.

*: crude: Model adjusted for age, sex.

‡: Model adjusted for age, sex, diabetes risk score, physical activity, total energy intake and saturated fat intake, Fiber, and Smoking.

a: GDQS score components: Healthy + Unhealthy + Unhealthy in excessive amount.

b: Food groups of healthy components of GDQS include Citrus fruits, Deep orange fruits, Other fruits, Dark green leafy vegetables, Cruciferous vegetables, Deep orange vegetables, Other vegetables, Legumes, Deep orange tubers, Nuts and seeds, Whole grains, Liquid oils, Fish and shellfish, Poultry and game meat, Low-fat dairy, Eggs (This component was categorized based on quartiles of this score).

c: Food groups of unhealthy components of GDQS include Processed meat, Refined grains, and baked goods, Sweets and ice cream, Sugar-sweetened beverages, juice, White roots, and tubers, Purchase deep fried foods (This component was categorized based on quartiles of this score).

d: Unhealthy excessive amount component of GDQS was categorized based on the median score (Due to the range of scores, it was not possible to make a quartile).

e: Food groups of unhealthy excessive amount components of GDQS include High-fat dairy, Red meat.

f: Total unhealthy score = Unhealthy components of GDQS sore + Unhealthy in excessive amount of GDQS.

amount food groups of the GDQS were conversely associated with T2D incidence in two crude and adjusted models [HR: 1, 0.89 (0.84–0.94). Finally, the sum of unhealthy and unhealthy in excessive amounts were conversely associated with T2D incidence in two crude and adjusted models [HR: 1, 0.86 (0.80–0.92), 0.93 (0.86–1.01), 0.89 (0.81–0.98) P trend = 0.009].

Fig 3 shows Cox proportional hazard regression plot for T2D according to quartiles of unhealthy component food group of the global diet quality score.

Participants in the fourth quartile received the highest score in the healthy food group from whole grains, other fruits, other vegetables, and citrus fruits, and in the unhealthy food group received the highest score from sugar-sweetened beverages, high-fat dairy, and red meat (S2 Table).

## Discussion

We found an inverse association between the GDQS and the risk of T2D incidence among Tehranian adults. Our findings indicate that healthy eating can play a significant role in preventing and managing T2D.

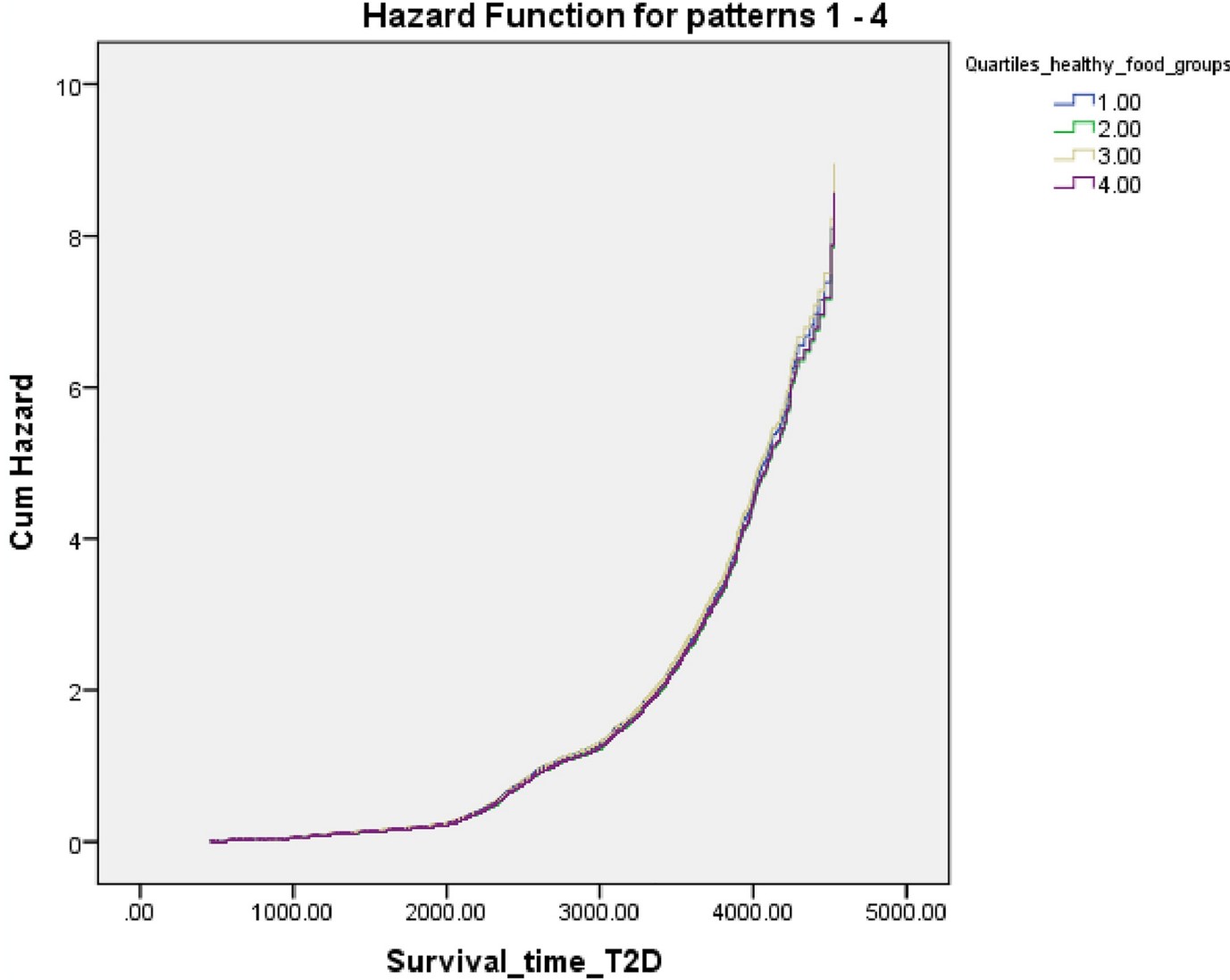

**Fig 2. Cox proportional hazard regression plot for type 2 diabetes (T2D) according to quartiles of healthy component food group of the global diet quality score.**

GDQS is a novel index that provides a more detailed breakdown of food group intake compared to similar indices. Limited studies available; therefore, we have used studies with similar topics here. Based on our knowledge, the association between GDQS and T2D has been studied only in the American population by Fang et al. in 2021. Our research was conducted in both male and female subjects in the Middle East, making this the first study to investigate this relationship in a developing country. Fung et al. indicated that a higher GDQS is associated with lower T2D risk, mostly due to a lower dietary intake of unhealthy foods in the US women [4]. In our analysis, high intakes of healthy food groups of the GDQS and low intakes of unhealthy in excessive amount food groups (high fat dairy and red meat) were strongly associated with a lower T2D risk; moreover, a higher score of the total unhealthy food groups of the GDQS or less consumption of unhealthy food groups, appeared to be negatively associated

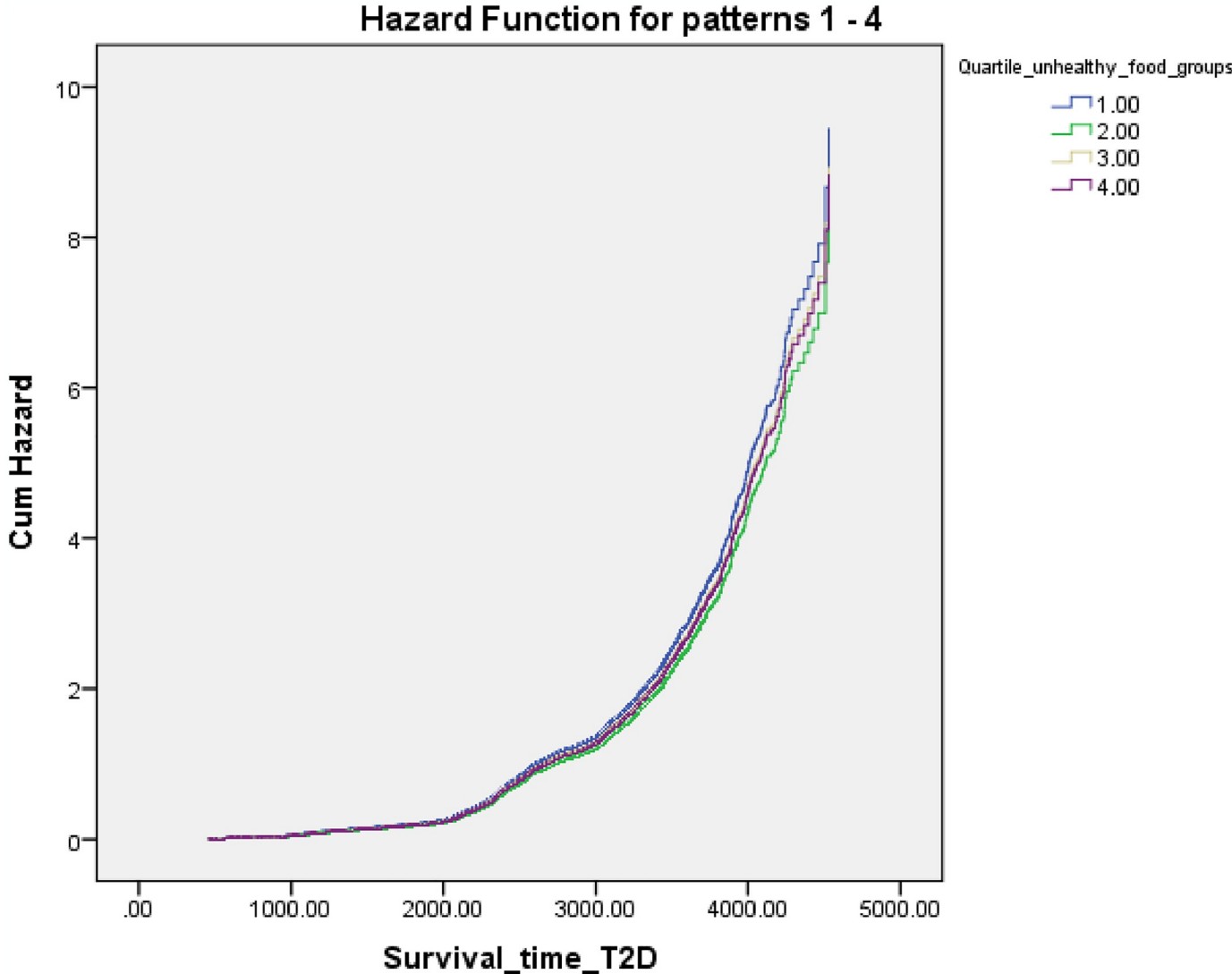

**Fig 3. Cox proportional hazard regression plot for type 2 diabetes (T2D) according to quartiles of unhealthy component food group of the global diet quality score.**

with T2D risk. It seems that the main negative role in the unhealthy subgroup is played by less consumption of red meat and high-fat dairy products.

The research by He et al. indicates that there was an inverse association between a high GDQS and inadequacy of nutrients, as well as metabolic syndrome among the Chinese adults. They showed that individuals in the highest quintile of GDQS demonstrated a 21% lower probability of developing metabolic syndrome compared to those in the lowest GDQS group [8]. In a prospective study by Angulo *et al*, an increase in the GDQS, was associated with lower weight and WC gain in the Mexican women [9]. In Kang *et al* study GDQS was positively associated with maternal mid-upper arm circumference and BMI among pregnant women in rural Ethiopia, however, the result did not indicate any relationship with overweight or diet-related morbidity [7]. Several investigations have demonstrated that eating a healthy diet reflects higher diet quality indices, which are associated with a lower risk of T2D in Asia [19, 20],

Europe [21], and the United States [22]. These studies used different diet quality indices, including the Healthy Eating Index (HEI) 2015, the Healthy Diet Score, the Alternate Healthy Eating Index (AHEI 2010), and the Mediterranean diet score. All mentioned scores recommend consuming more whole grains, fruits, vegetables, lean protein, and less sugar, red meat, processed meat, and refined grains in general. The sum of food groups ranged from 6 (Healthy Nordic Food Index) to 11 (AHEI) and 13 (HEI-2015). Like other diet quality indices, the GDQS index comprises healthy and unhealthy food groups; however, the GDQS food groups are classified more detail and form 25 groups.

Diabetes is inversely associated with healthy dietary patterns, similar to the healthy subgroup of GDQS [23]. The previous studies showed inverse association between the Mediterranean diet score and T2D incidence [24–26]. Also, Chen et al. demonstrated that adherence to a high-quality diet including the alternate Mediterranean diet (aMED), AHEI-2010, the Dietary Approaches to Stop Hypertension (DASH) diet, an overall plant-based diet index, and a healthful plant-based diet index was significantly associated with a lower risk of T2D in an Asian population [27].

On the other hand, there was no association between HEI-2015 and the risk of T2D [28, 29]. In the meta-analysis by Schwingshackl et al, the association between food groups and T2D was investigated; the result showed that selecting specific optimal intakes of specific food groups, such as whole grains, vegetables, fruits, and dairy; and reducing red and processed meats, sugar-sweetened beverages and eggs, can significantly reduce the risk of T2D [30].

The other meta-analysis by Wang et al, recently reported that green leafy vegetables, wholegrain, and cereal fiber consumption were associated with lower diabetes risks, also the results indicated that a higher intake of fruits, especially berries, cruciferous vegetables, yellow vegetables or their fiber is associated with a lower risk of T2D [31].

Regarding fruits, recent research suggests that 200 grams of fruits a day may help prevent T2D, and fruits with low glycemic loads may help diabetic patients control their blood sugar levels [32]. Schwingshackl et al suggest that olive oil intake is associated with the prevention and management of T2D [30], but nuts were not linked to diabetes risk [31, 33]. Among the unhealthy food groups, high intakes of potatoes especially French fries, and sugar-sweetened beverages, had shown previously to be positively related to a higher risk of T2D [4, 30, 34]. The results of a recent meta-analysis by Sanders *et al*, showed that there is no association between red meat consumption with glycemia and insulinemic risk factors of T2D; however, they recommend further research on other markers of glucose homeostasis [35]. In contrast with Sander's study, another meta-analysis indicated that both processed and unprocessed red meat are certainly associated with T2D incidence [36].

Regarding dairy products, Tong et al. reported a positive association between dairy intake and T2D development, supporting the beneficial effects of dairy consumption on controlling T2D [37]. According to Alvarez-Bueno et al, dairy consumption, particularly low-fat dairy, and yogurt (80–125 gr/day), is associated with a lower risk of T2D [38], even so, based on Moslehi et al's study, there was no association between high-fat and low-fat dairy products and T2D [1], which is in line with Morato-Martínez et al's study [39]. In our study there was a significant association between high consumption of high-fat dairy products and red meat with diabetes incidence, it appears that low intake of high-fat dairy products can contribute to a lower incidence of T2D, However, there is still insufficient evidence on high-fat dairy, and more research is needed to distinguish between fat levels in dairy products (high or low) and sweetener levels in milk and yogurt.

Our investigation had some strength to mention. First, the urban population-based TLGS contains longitudinal data. Second, we assessed and adjusted probable confounders on the incidence of T2D. Third, the definition of diabetes was determined according to fasting blood

glucose experiments not self-reporting. Fourth, the calculation for the nutrients and food groups was conducted cumulatively through the follow-up surveys. Finally, in this study, we analyzed unhealthy and unhealthy in excessive amounts components of GDQS separately and also analyzed both groups in combination.

Our study had several limitations to mention. As our study did not comprise data on the common techniques used for cooking or preparation of foods, so we could not assess this effect on the progression of T2D. In addition, even though we tried to detect all confounding variables, some residual confounding might still exist due to the absence of measurement or knowledge. Dietitians collect data about dietary intake without replicas of foods. More studies are needed for generalizability in different populations.

## Conclusion

The findings of this research confirm that following a dietary pattern in line with this global easily practical diet metric reduces the risk of T2D in Tehranian adults. Our results disclosed that higher intake of healthy food groups of GDQS and lower intake of unhealthy food groups of GDQS were associated with a lower risk of T2D incidence. Moreover, it appears that the most adverse impact of unhealthy food categories is tied to lower intake of high-fat dairy products and red meat, although it is essential to be mindful of the other food groups within the unhealthy category. Finally, in our population, this index can be used to evaluate dietary intakes in primary assessments for the prevention of T2D. Examining this index in different regions of Iran with high prevalence of diabetes or other consequences will be beneficial to create a national index and guideline. In addition, with the help of this index, the nutritional status can be compared with other parts of the world.

## Supporting information

**S1 Table. Food group components, classification, and point values of the global diet quality score (GDQS).**
(DOCX)

**S2 Table. The number (percentage) of participants according to quartiles of food group scores and based on quartiles of global diet quality score (GDQS).**
(DOCX)

## Acknowledgments

The author would be grateful to the participants and the TLGS personnel for their participation.

## Author Contributions

**Formal analysis:** Shahrzad Daei, Firoozeh Hosseini-Esfahani.

**Investigation:** Azam Ildarabadi.

**Project administration:** Parvin Mirmiran.

**Supervision:** Firoozeh Hosseini-Esfahani, Parvin Mirmiran, Fereidoun Azizi.

**Validation:** Firoozeh Hosseini-Esfahani, Parvin Mirmiran.

**Writing – original draft:** Shahrzad Daei.

**Writing – review & editing:** Firoozeh Hosseini-Esfahani, Azam Ildarabadi.

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
