## [Decision Letter · Decision Letter 0]

15 Jul 2024

PONE-D-24-19056Associations between the Global Diet Quality Score and risk of Type 2 Diabetes: Tehran Lipid and Glucose StudyPLOS ONE

Dear Dr. Hosseini-Esfahani,

Thank you for submitting your manuscript to PLOS ONE. After careful consideration, we feel that it has merit but does not fully meet PLOS ONE’s publication criteria as it currently stands. Therefore, we invite you to submit a revised version of the manuscript that addresses the points raised during the review process.

plosone@plos.org. Please submit your revision by Aug 29 2024 11:59PM , log on to https://www.editorialmanager.com/pone/ and select the 'Submissions Needing Revision' folder to locate your manuscript file.

Please include the following items when submitting your revised manuscript:A rebuttal letter that responds to each point raised by the academic editor and reviewer(s). You should upload this letter as a separate file labeled 'Response to Reviewers'.A marked-up copy of your manuscript that highlights changes made to the original version. You should upload this as a separate file labeled 'Revised Manuscript with Track Changes'.An unmarked version of your revised paper without tracked changes. You should upload this as a separate file labeled 'Manuscript'.

We look forward to receiving your revised manuscript.

Kind regards,

Meisam Akhlaghdoust, M.D., M.P.H.

Academic Editor

PLOS ONE

https://www.researchgate.net/publication/363627658_Diet_quality_indices_and_the_risk_of_type_2_diabetes_in_the_Tehran_Lipid_and_Glucose_Study

https://www.frontiersin.org/articles/10.3389/fnut.2022.891111/full

In your revision ensure you cite all your sources (including your own works), and quote or rephrase any duplicated text outside the methods section. Further consideration is dependent on these concerns being addressed.

 [This work was supported by the Research Institute for Endocrine Science, Shahid Beheshti University of Medical Science (Tehran, Iran) under Grant number. 43003733].  

5. In the online submission form you indicate that your data is not available for proprietary reasons and have provided a contact point for accessing this data. Please note that your current contact point is a co-author on this manuscript. According to our Data Policy, the contact point must not be an author on the manuscript and must be an institutional contact, ideally not an individual. Please revise your data statement to a non-author institutional point of contact, such as a data access or ethics committee, and send this to us via return email. Please also include contact information for the third party organization, and please include the full citation of where the data can be found.

Additional Editor Comments (if provided):

Reviewers' comments:

Reviewer's Responses to Questions

**Comments to the Author**

1. Is the manuscript technically sound, and do the data support the conclusions?

Reviewer #1: Yes

Reviewer #2: Yes

2. Has the statistical analysis been performed appropriately and rigorously? 

Reviewer #1: Yes

Reviewer #2: Yes

3. Have the authors made all data underlying the findings in their manuscript fully available?

Reviewer #1: No

Reviewer #2: No

4. Is the manuscript presented in an intelligible fashion and written in standard English?

Reviewer #1: Yes

Reviewer #2: Yes

5. Review Comments to the Author

Reviewer #1: I would like to thank the authors to design this valuable study. However, there are some comments as follows:

Please revise the manuscript in terms of English language and grammar.

Abstract:

Background: Please use association instead of relation. Please revise it in the whole manuscript.

Methods: In this secondary analysis, we included elective adult participants (n=5948) from 28 the Tehran Lipid and Glucose Study. Please mention the Phase of the TLGS you used the data.

Please mention the components of GDQS score.

Results:

Please report mean(SD) instead of mean±SD

Please report the 6 year risk of T2DM through K-M method, as well.

Line 36: The healthy components of

Line 37: 1 for what?? I think it is 0.91

Line 39: 1, 0.86� 0.86 0.93

Please recheck all the numbers you reported.

Keywords: please write them based on MeSH database.

Methods:

This section has been written well. However, please write this section according to the STROBE writing standard in terms of headings and subheadings.

Results:

Please report mean(SD) instead of mean±SD and consider in the whole manuscript.

Please report the 6 year risk of T2DM through K-M method, as well.

Line 228-233: please recheck the numbers you reported in this paragraph.

Reviewer #2: Comments for the authors:

Thank you for your efforts. There are some major points that need improvement:

Abstract

1.The background of abstract provides no background, but just objective of the study.

Introduction

1.Line 48, the sentence “with a T2D incidence rate of 36.3% per 1000 person-years”: Incidence rates are typically expressed as the number of new cases per person-time (e.g., per 1000 person-years), not as a percentage. Percentages are usually used to express prevalence or risk, not incidence rates.

2.Line 54, the sentence “Current meta-analysis studies suggest”: You are talking about studies while citing only one.

3.While the GDQS is introduced, a bit more detail on its composition and why it is considered a robust measure would be beneficial.

4.The examples of Ref. 7&9 are not related that much to the topic, as of different study population. You must provide relevant example to highlight result of similar studies.

5.The transition from definitions and known areas to the specific focus of the current study could be smoother, emphasizing the novelty, the unknown aspects and importance of the research.

Method

1.Not a common format to start the method section by ethics statement. Please provide details of study design.

2.Line 77, the sentence “this study was set up in district no.13 of Tehran to prevent non-communicable disease risk factors and their consequences”: I do not think that the study was directly designed to prevent NCDs.

3.How did you choose to select 8048 participants? Please also explain more about randomization.

4.You mentioned that total of 7268 individuals were selected as the baseline population, of whom 597 participants were excluded. Finally, you mentioned 5948 subjects remained for analysis, which is not logical. Also, please recheck all numbers mentioned in Figure 1. The summation of numbers included and excluded must be rechecked.

5.I do not understand why the manuscript “Issues in analysis and presentation of dietary data” should be cited in the method section while explaining about inclusion of cases of your study (Ref. 10). I also doubt the necessity of citing your previous published work (Ref. 11&12) in method section.

6.Line 100: Please define USDA.

Result

1.Better to include percentage of each frequency.

2.Consider providing more detailed subgroup analyses. For Table 1, post hoc analysis are recommended.

3.Include p-values for the differences in baseline characteristics to demonstrate statistical significance.

4.The approach to handling missing data is not discussed.

Discussion

Overall discussion section lacks logical flow and smooth transitions.

1.Line 256, the sentence “It seems that this association can help us to consume proper healthy food groups and limit unhealthy foods in excessive amounts”: Please provide more details, how it seems to help. Not a good beginning for discussion. Please elaborate more on the important findings and then turn to the application.

2.The third paragraph of discussion contains listing many references that have used GDQA index without providing any comparison with your results.

3.Line 262: I do not understand why the sentence begins with “however”. What contrast dose it show?

4.Line 274&276: The reference 8 should be mentioned as He et al. not Fang et al.

5.Line 273-5, the sentence “Based on our knowledge the relationship between GDQS and T2D has been studied only in the American population by Fang et al in 2021..”: This is incorrect. Other references:

https://doi.org/10.1093/jn/nxab195

https://doi.org/10.1093/cdn/nzaa061_029

https://doi.org/10.1093/aje/kwy183

6.Lines 283-290: Lacks coherence and smooth transitions between different indexes.

7.The discussion could benefit from a more detailed comparison of GDQS with other indices, specifically highlighting why GDQS might be more effective or accurate.

8.Lines 295-304: Seems more like dietary recommendation for diabetics, rather than discussion.

9.Provide clearer public health implications if the results.

10.In limitation, mention the generalizability and the need for further studies in diverse populations.

Thank you for addressing these points in advanced. I look forward to seeing the revised manuscript.

6. PLOS authors have the option to publish the peer review history of their article (what does this mean?). If published, this will include your full peer review and any attached files.

Reviewer #1: **Yes: **Mahin Nomali, Epidemiologist

Reviewer #2: **Yes: **Noosha Samieefar

---

## [Author Response · Author response to Decision Letter 0]

19 Sep 2024

Dear Dr. Meisam Akhlaghdoust

I would like to express my gratitude for your valuable time and thoughtful comments on our manuscript “Associations between the Global Diet Quality Score and risk of Type 2 Diabetes: Tehran Lipid and Glucose Study”. Your insights have been instrumental in refining our work.

In response to your suggestions, we have diligently revised the manuscript. To facilitate your review, we have provided a point-by-point summary of the requested changes, and the modifications are highlighted in the submitted file. We are hopeful that the revised manuscript now aligns with the standards for publication in “PLOS ONE” Your interest in our research and the opportunity to enhance its quality are greatly appreciated. We remain open to any additional feedback or further revisions that you may recommend ensuring the manuscript meets the journal’s criteria .Once again, thank you for your commitment to advancing scientific discourse, and we look forward to your feedback on the revised version.

Please find the below comments/questions by the editor/reviewers (in black color) and our response/clarification/explanation to the comments and questions (in blue color).

Best regards,

Parvin Mirmiran, Prof.

E-mail: parvin.mirmiran@sbmu.ac.ir

Parvin.mirmiran@gmail.com

Reviewer #1:

 I would like to thank the authors to design this valuable study. However, there are some comments as follows:

Please revise the manuscript in terms of English language and grammer.

Abstract:

Background: Please use association instead of relation. Please revise it in the whole manuscript.

Reply: Agreed and corrected in all of the manuscript.

Methods: In this secondary analysis, we included elective adult participants (n=5948) from the Tehran Lipid and Glucose Study. Please mention the Phase of the TLGS you used the data.

Reply: Agreed and corrected

In this secondary analysis, we included elective adult participants (n=5948) from the third and fourth phase of the Tehran Lipid and Glucose Study. Participants checked out until the sixth phase with an average follow-up of 6.65 years.

Please mention the components of GDQS score. 

Reply: The GDQS were calculated including healthy and unhealthy food group scores.

Food groups are divided into 16 healthy including kinds of fruits and vegetables, legumes, nuts and seeds, whole grain, fish, poultry, liquid oil, low-fat dairy, and egg; 7 unhealthy including processed meats, refined grains, sweets, sugar-sweetened beverages, potato or cassava flour, juice, and deep fried foods; and two unhealthy when consumed more than recommendation including red meat and high-fat dairy. (page 4, line 61-64)

Results:

Please report mean(SD) instead of mean±SD

Reply: Agreed and corrected

Please report the 6 years risk of T2DM through K-M method, as well.

Reply: Diagrams were added at the end of the figure.

Line 36: The healthy components of 

Reply: A higher score is given to the consumption of healthy food groups.

Line 37: 1 for what?? I think it is 0.91 

Reply: Number one is our reference.

Line 39: 1, 0.86� 0.86 0.93

Reply: The number 93 was changed to 0.93 and corrected.

Please recheck all the numbers you reported.

Reply: Agreed and corrected.

Keywords: please write them based on MeSH database.

Reply: Agreed and corrected.

Methods:

This section has been written well. However, please write this section according to the STROBE writing standard in terms of headings and subheadings.

Reply: Agreed and corrected.

Results:

Line 228-233: please recheck the numbers you reported in this paragraph.

Reply: Agreed and corrected.

The score of unhealthy in excessive amount food groups of the GDQS were conversely associated with T2D incidence in two crude and adjusted models [HR: 1, 0.89 (0.84-0.94). (page 14 line 237)

Reviewer #2:

 Comments for the authors:

Thank you for your efforts. There are some major points that need improvement:

Abstract

1.The background of abstract provides no background, but just objective of the study.

Reply: Agreed and corrected.

Previous studies reported that focusing on healthy lifestyle especially high dietary quality is necessary for preventing type 2 diabetes (T2D).

Introduction

1. Line 48, the sentence “with a T2D incidence rate of 36.3% per 1000 person-years”: Incidence rates are typically expressed as the number of new cases per person-time (e.g., per 1000 person-years), not as a percentage. Percentages are usually used to express prevalence or risk, not incidence rates.

Reply: Following your comments this part has been revised.

In the Iranian population, there are >800,000 new cases of T2D per year, with a T2D incidence rate of 36.3 per 1000 person-years [3].( page 4 line 50-51)

2.Line 54, the sentence “Current meta-analysis studies suggest”: You are talking about studies while citing only one.

Reply: Agreed and corrected.

3.While the GDQS is introduced, a bit more detail on its composition and why it is considered a robust measure would be beneficial. 

Reply: Food groups are divided into 16 healthy including kinds of fruits and vegetables, legumes, nuts and seeds, whole grain, fish, poultry, liquid oil, low fat dairy and egg; 7 unhealthy including processed meats, refined grains, sweets, sugar sweetened beverages, potato or cassava flour, juice, and deep-fried foods; and 2 unhealthy when consumed more than recommendation including red meat and high fat dairy. Page 4 line 61-65

4.The examples of Ref. 7&9 are not related that much to the topic, as of different study population. You must provide relevant example to highlight result of similar studies. 

Reply: The GDQS index is a new measure with limited studies available. Since there are no research articles specifically examining this index in relation to diabetes, we relied on studies focused on related indices and diabetes occurrence.

Method

1.Not a common format to start the method section by ethics statement. Please provide details of study design.

Reply: Following your comment, this section has been revised.

The Tehran lipid and glucose study (TLGS) was performed prospectively on the 13th district of Tehran (the capital of Iran) residents to identify risk factors for non-communicable diseases [10, 11]. According to this study, the initial sample of 15005 participants (aged ≥3 years) were enrolled between 1999 and 2001. During the study, participants were followed every three years: wave 2 (2002-2005), wave 3 (2005-2008), wave 4 (2008-2011), wave 5 (2012-2015), wave 6 (2015-2018) to amend their demographic and health-related data, as well as identified diseases that have recently emerged. For this secondary analysis of subjects participating in waves 3 and 4 (baseline of our study), 8048 individuals aged ≥18 years were randomly selected to complete the dietary assessment based on age and sex distribution. Subjects with over- or under-reporting of energy intake (≥4,200 or <800 kcal/day) (n=780) were excluded [12], and a total of 7268 individuals with accessible biochemical, anthropometric, and dietary data were entered as the baseline population. They were tracked until phase 6; subjects who entered phase 3 or 4 in this study were respectively followed three and twice for the outcome measurement. Among these subjects, pregnant or lactating women, participants with T2D diagnosis based on fasting blood glucose (FBG) measurements or self-reported use of glucose-lowering medications at baseline (n=597), and subjects with missing data (n=208) were excluded. Also, subjects who did not provide follow-up data were excluded (n=515) from the study participants. As a result, 5948 subjects remained for analysis (Page 5-6 line 76-93).

2.Line 77, the sentence “this study was set up in district no.13 of Tehran to prevent non-communicable disease risk factors and their consequences”: I do not think that the study was directly designed to prevent NCDs.

Reply: Revisions were made to improve clarity in this section.

3.How did you choose to select 8048 participants? Please also explain more about randomization.

Reply: Revisions were made to improve clarity in this section.

The Tehran lipid and glucose study (TLGS) was performed prospectively on the 13th district of Tehran (the capital of Iran) residents to identify risk factors for non-communicable diseases [10, 11]. According to this study, the initial sample of 15005 participants (aged ≥3 years) were enrolled between 1999 and 2001. During the study, participants were followed every three years: wave 2 (2002-2005), wave 3 (2005-2008), wave 4 (2008-2011), wave 5 (2012-2015), wave 6 (2015-2018) to amend their demographic and health-related data, as well as identified diseases that have recently emerged. For this secondary analysis of subjects participating in waves 3 and 4 (baseline of our study), 8048 individuals aged ≥18 years were randomly selected to complete the dietary assessment based on age and sex distribution. Subjects with over- or under-reporting of energy intake (≥4,200 or <800 kcal/day) (n=780) were excluded [12], and a total of 7268 individuals with accessible biochemical, anthropometric, and dietary data were entered as the baseline population. They were tracked until phase 6; subjects who entered phase 3 or 4 in this study were respectively followed three and twice for the outcome measurement. Among these subjects, pregnant or lactating women, participants with T2D diagnosis based on fasting blood glucose (FBG) measurements or self-reported use of glucose-lowering medications at baseline (n=597), and subjects with missing data (n=208) were excluded. Also, subjects who did not provide follow-up data were excluded (n=515) from the study participants. As a result, 5948 subjects remained for analysis (Page 5-6 line 76-93).

4.You mentioned that total of 7268 individuals were selected as the baseline population, of whom 597 participants were excluded. Finally, you mentioned 5948 subjects remained for analysis, which is not logical. Also, please recheck all numbers mentioned in Figure 1. The summation of numbers included and excluded must be rechecked.

Reply: Revisions were made to improve clarity in this section.

5.I do not understand why the manuscript “Issues in analysis and presentation of dietary data” should be cited in the method section while explaining about inclusion of cases of your study (Ref. 10). I also doubt the necessity of citing your previous published work (Ref. 11&12) in method section. 

Reply: Agreed and corrected 

6.Line 100: Please define USDA. 

Reply: The complete name added.

Due to the incompleteness and limited data on the nutrient content of cooked food items in the Iranian Food Composition Table (FCT), United States Department of Agriculture (USDA) data was used. The Iranian FCT was applied for national foods that could not be incorporated into the USDA portion size [2]. (Page 6 line 100-106)

Result

1.Better to include percentage of each frequency.

Reply: Modifications were implemented.

2.Consider providing more detailed subgroup analyses. For Table 1, post hoc analysis are recommended.

Reply: Following your comments post hoc analysis added (Page 12 line 105).

3. Include p-values for the differences in baseline characteristics to demonstrate statistical significance.

Reply: Modifications were implemented.

4. The approach to handling missing data is not discussed.

Reply: Individuals with missing data were excluded. 

Discussion

Overall discussion section lacks logical flow and smooth transitions.

1.Line 256, the sentence “It seems that this association can help us to consume proper healthy food groups and limit unhealthy foods in excessive amounts”: Please provide more details, how it seems to help. Not a good beginning for discussion. Please elaborate more on the important findings and then turn to the application. 

Reply: Revisions were made to improve clarity in this section.

We found an inverse association between the GDQS and the risk of T2D incidence among Tehranian adults. Our findings indicate that healthy eating can play a significant role in preventing and managing T2D. GDQS is a novel index that provides a more detailed breakdown of food group intake compared to similar indices. Limited studies available; therefore, we have used studies with similar topics here. Based on our knowledge, the association between GDQS and T2D has been studied only in the American population by Fang et al. in 2021. Our research was conducted in both male and female subjects in the Middle East, making this the first study to investigate this relationship in a developing country. Fang et al. indicated that a higher GDQS is associated with lower T2D risk, mostly due to a lower dietary intake of unhealthy foods in the US women [4]. In our analysis, high intakes of healthy food groups of the GDQS and low intakes of unhealthy in excessive amount food groups (high fat dairy and red meat) were strongly associated with a lower T2D risk; moreover, a higher score of the total unhealthy food groups of the GDQS or less consumption of unhealthy food groups, appeared to be negatively associated with T2D risk. It seems that the main negative role in the unhealthy subgroup is played by less consumption of red meat and high-fat dairy products. The research by He et al. indicates that there was an inverse association between a high GDQS and inadequacy of nutrients, as well as metabolic syndrome among the Chinese adults. They showed that individuals in the highest quintile of GDQS demonstrated a 21% lower probability of developing metabolic syndrome compared to those in the lowest GDQS group [8]. In a prospective study by Angulo et al, an increase in the GDQS, was associated with lower weight and WC gain in the Mexican women [9]. In Kang et al study GDQS was positively associated with maternal mid-upper arm circumference and BMI among pregnant women in rural Ethiopia, however, the result did not indicate any relationship with overweight or diet-related morbidity [7] page 14,15 line 261-285).

2.The third paragraph of discussion contains listing many references that have used GDQS index without providing any comparison with your results. 

Reply: There are limited studies available; therefore, we have used studies with similar topics here.

3.Line 262: I do not understand why the sentence begins with “however”. What contrast dose it show?

Reply: Thanks for your comment. I discussed various healthy eating indexes (HEI, Mediterranean diet, etc.) with different score calculations. Ultimately, their patterns are similar, and we can compare them to some extent.

4.Line 274&276: The reference 8 should be mentioned as He et al. not Fang et al 

Reply Modifications were implemented.

5.Line 273-5, the sentence “Based on our knowledge the relationship between GDQS and T2D has been studied only in the American population by Fang et al in 2021..”: This is incorrect. Other references:

https://doi.org/10.1093/jn/nxab195

https://doi.org/10.1093/cdn/nzaa061_029

https://doi.org/10.1093/aje/kwy183

Reply: The first and second suggested references are the same and are included in the study. The third reference pertains to diet quality indices that differ by GDQS. It was a helpful article, and based on your suggestion, we have included it in our article.

Also, Chen et al. demonstrated that adherence to a high-quality diet including the alternate Mediterranean diet (aMED), AHEI-2010, the Dietary Approaches to Stop Hypertension (DASH) diet, an overall plant-based diet index, and a healthful plant-based diet index was significantly associated with a lower risk of T2D in an Asian population [28] (page 19,20 line 296-299).

6.Lines 283-290: Lacks coherence and smooth transitions between different indexes. 

Reply: The discussion section has been revised.

7.The discussion could benefit from a more detailed comparison of GDQS with other indices, specifically highlighting why GDQS might be more effective or accurate. 

Reply: GDQS is a novel index that provides a more detailed breakdown of food group intake compared to other similar indices.

8.Lines 295-304: Seems more like dietary recommendation for diabetics, rather than discussion.

Reply: Thank you f

---

## [Decision Letter · Decision Letter 1]

4 Nov 2024

Associations between the Global Diet Quality Score and risk of Type 2 Diabetes: Tehran Lipid and Glucose Study

PONE-D-24-19056R1

Dear Dr. Hosseini-Esfahani,

We’re pleased to inform you that your manuscript has been judged scientifically suitable for publication and will be formally accepted for publication once it meets all outstanding technical requirements.

Kind regards,

Meisam Akhlaghdoust, M.D., M.P.H.

Academic Editor

PLOS ONE

Additional Editor Comments (optional):

Reviewers' comments:

Reviewer's Responses to Questions

**Comments to the Author**

1. If the authors have adequately addressed your comments raised in a previous round of review and you feel that this manuscript is now acceptable for publication, you may indicate that here to bypass the “Comments to the Author” section, enter your conflict of interest statement in the “Confidential to Editor” section, and submit your "Accept" recommendation.

Reviewer #2: All comments have been addressed

2. Is the manuscript technically sound, and do the data support the conclusions?

Reviewer #2: Yes

3. Has the statistical analysis been performed appropriately and rigorously? 

Reviewer #2: Yes

4. Have the authors made all data underlying the findings in their manuscript fully available?

Reviewer #2: No

5. Is the manuscript presented in an intelligible fashion and written in standard English?

Reviewer #2: Yes

6. Review Comments to the Author

Reviewer #2: Thanks for your efforts in preparing the revision.

7. PLOS authors have the option to publish the peer review history of their article (what does this mean?). If published, this will include your full peer review and any attached files.

Reviewer #2: **Yes: **Noosha Samieefar

---

## [Editor Report · Acceptance letter]

11 Nov 2024

PONE-D-24-19056R1 

PLOS ONE

Dear Dr. Hosseini-Esfahani, 

I'm pleased to inform you that your manuscript has been deemed suitable for publication in PLOS ONE. Congratulations! Your manuscript is now being handed over to our production team.

Kind regards, 

on behalf of

Dr. Meisam Akhlaghdoust 

Academic Editor

PLOS ONE